# The Correlation between the Structure Characteristics and Gasification Characteristics of Tar Residue from Pyrolysis

**Jiahui Li [1], Weiguo Li [2], Xuefeng She [1,\*], Jianhong Shi [2], Peifang Lin [2] and Qingguo Xue [1]**

[1] State Key Laboratory of Advanced Metallurgy, University of Science and Technology Beijing, Beijing 100083, China
[2] Guangdong Huaxin Environmental Protection Technology Co., Ltd., Guangzhou 512199, China
\* Correspondence: shexuefeng@ustb.edu.cn

**Abstract:** Pyrolysis is an efficient method for utilizing tar residue as a resource, and the structural properties of tar residue from pyrolysis (TRP) significantly impact subsequent gasification. The study examines the changes in the microscopic morphology, surface area, and carbon structure characteristics of TRPP as a function of pyrolysis temperature to elucidate the influence of pyrolysis temperature on the $CO_2$ gasification characteristic parameters of TRP. Additionally, the investigation explores the relationship between surface structure and carbon structure characteristic parameters and gasification parameters at various stages. The findings indicated that the surface morphology of TRP synthesized at different pyrolysis temperatures (500–900 °C) was divided into two stages: the development of pores and the jamming of pores. With increasing pyrolysis temperature, the bigger aromatic nucleus was formed in the TRP without complete graphitization, and more amorphous carbon was consumed. TRP prepared at a pyrolysis temperature of 700 °C had the best gasification reactivity. By combining XRD, Raman, and gas adsorption techniques, the correlations between the surface structure and carbon structure parameters and the gasification characteristic parameters were established to evaluate the main factors influencing the gasification reaction. In the early stage of the gasification reaction, the carbon structure played a more important role than the surface structure. As the gasification reaction proceeded, the relationship between the surface structure and the gasification reaction was closer.

**Keywords:** tar residue; pyrolysis temperature; structure characteristics; gasification characteristics



## 1. Introduction

Coal tar residue is formed during the coal gasification and coking process when organic compounds with high boiling points condense and are mixed with solid particles entrained in the gas. The main components are tar, pulverized coal, coke powder, and organic substances such as polycyclic aromatic hydrocarbons, benzene, and phenols [1]. According to data from the International Energy Agency (IEA), the global output of coke was approximately 706 million tons in 2021. China is the largest producer of coke, accounting for more than half of the global market, with an annual production of approximately 400 million tons. Other major producers of coke include India, Japan, Russia, and the United States. The proportion factor of tar residue production is approximately one thousandth, so a large amount of untreated tar residue will be generated globally every year. Conventional treatment methods in the form of direct use as fuel or disposal generate toxic PAHs (polycyclic aromatic hydrocarbons), which can have adverse effects on the ecological environment and human health [2–4]. At present, tar residue is used mainly for the coking of tar residue mixed with coal [5], causing fluctuations in coke quality and increasing the heat load of the coke oven [6]. As a high calorific value carbonaceous material [7], determining how to properly and effectively utilize tar residue has become a pressing issue in recent years.

Pyrolysis is the thermal decomposition of tar residue under hypoxic or anoxic conditions into combustible gas, oil, and coke for secondary use as fuel or other chemical materials. There have been numerous studies on the pyrolysis of tar residue, which can be divided into three stages: drying and degassing, active thermal decomposition, and pyrolytic condensation [8] (where the condensation reaction can be essentially completed at 900 °C) [9]. In pyrolysis, carbon monoxide, carbon dioxide, methane, and other gases will be released [10]. The final temperature and particle size of pyrolysis greatly influence the pyrolysis yield. Lu et al. [1] investigated the pyrolytic characteristics of the coal tar residue and its extraction residue (RCTR) with organic solvents. The extractant type affected the activation energy and reactivity of RCTR.

Prior research has primarily concentrated on scrutinizing either the pyrolysis mechanism or the extracted output of tar residues. However, there is a paucity of studies examining the characteristics of pyrolysis products, particularly the association between the structural features and gasification properties of tar residue obtained from slow pyrolysis, also known as TRP. Pyrolysis destroys hazardous components and reduces the volume and quantity of solid waste [11], while the remaining TRP contains fixed carbon, etc., which can be recycled, for example, as a reducing agent to be added to a rotary bottom furnace. As a result, studying the gasification properties of TRP prepared at various pyrolysis temperatures can aid in determining the optimal pyrolysis temperature for producing TRP with high gasification reactivity. It serves as a foundation for determining the pyrolysis temperature of the tar residue, which is critical for tar residue recycling and reuse, as well as the subsequent application of TRP gasification. This can not only alleviate the energy crisis and reduce the dependence on traditional fossil fuels but also improve energy utilization efficiency and reduce negative environmental impacts. Meanwhile, in the process of implementing this technology, it will also create employment opportunities and promote local economic development. In addition, the technology and process involved in the study of thermal decomposition and the subsequent gasification of tar slag also provide a basis for the treatment of other energy-containing solid wastes in the metallurgical industry, promoting sustainable utilization modes of energy and resources. Gasification characteristics are influenced by factors such as pore structure and carbon structure [12]. Wang et al. [13] analyzed the $CO_2$ gasification properties of three biomass chars and anthracite chars by the thermogravimetric analysis method and concluded that the gasification properties of the chars were determined by the carbon structure. Wu et al. [14] investigated the effect of cellulose on the physicochemical properties and gasification reactivity of co-pyrolysis char. They showed that the gasification reactivity index was exponentially related to the parameters of the microcrystalline structure. Wu et al. [15] investigated the gasification reactivity of coal char and petroleum coke prepared under different pyrolysis conditions. It was observed that the reactivity of these samples depends on their physical properties, such as pore structure. Huo et al. [16] compared the gasification reactivities and the pore structure characteristics of the feed coal and residues in industrial gasification plants. They found that the reactivity of the gasification of raw coal is the highest due to the largest surface area and the most developed pore structure.

This study examines TRP prepared from tar residue at different pyrolysis temperatures. The physicochemical structure of TRP was determined by the combined application of SEM, BET, XRD, and Raman spectroscopy. Non-isothermal thermogravimetric analysis was used to investigate the $CO_2$ gasification experiments of TRP and to examine the correlation between the surface structure and carbon structure characteristic parameters and the gasification characteristic parameters. In addition, the pyrolysis temperature of TRP was also determined with optimal gasification reactivity.

## 2. Experimental

### 2.1. Materials

The tar residue, obtained from a coking plant in China, was used as feedstock, and the approximate analysis (SDTGA6000A, Changsha, China) and ultimate analysis (Elementar-

UNICUBE, Frankfurt, Germany) data are listed in Table 1. From Table 1, it can be seen that the tar residue has the properties of high volatile matter and low ash content.

**Table 1.** Results of the proximate and ultimate analysis of the tar residue.

| Sample | Proximate Analysis/wt.% | | | | Ultimate Analysis/wt.% | | | | |
|---|---|---|---|---|---|---|---|---|---|
| | $M_{ad}$ | $A_{ad}$ | $V_{ad}$ | $FC_{ad}$ | C | O | H | N | S |
| tar residue | 5.51 | 5.23 | 43.31 | 45.95 | 80.32 | 12.7 | 3.27 | 1.33 | 0.76 |

Note: M—moisture; A—ash; V—volatile matter; FC—fix carbon.

### 2.2. TRP Preparation

As shown in Figure 1, a high-temperature vertical furnace reactor was used to prepare the TRP under a nitrogen atmosphere at the designed temperature. The heating rate of the resistance furnace can be controlled. In the experiment, a corundum crucible containing tar residue was positioned at the base of a quartz tube, which measured 80 mm in bottom diameter and 750 mm in height. A flanged valve with inlet and outlet ports was utilized to seal the end of the quartz tube. Prior to initiating the temperature protocol, $N_2$ gas was purged for 20 min to clean the quartz tube. A heating rate of 7 °C/min was applied for the pyrolysis of the tar residue in $N_2$, with a flow rate of 50 mL/min. After reaching the final designed temperature (500, 600, 700, 800, or 900 °C), the pyrolysis temperature was continued for 30 min and then cooled to room temperature under an $N_2$ atmosphere. Finally, part of the TRP was crushed into 50-mesh powder.

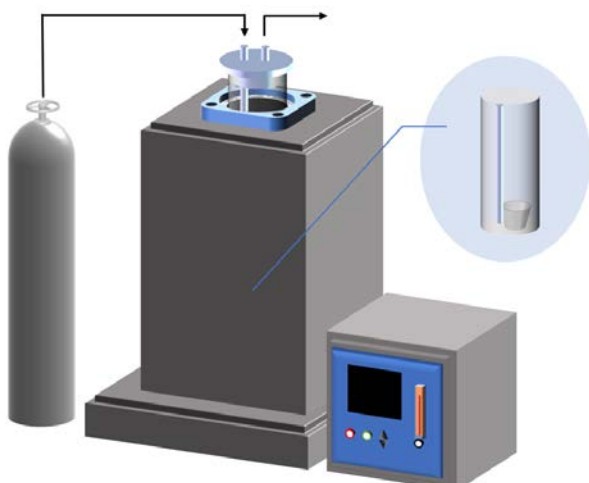

**Figure 1.** Schematic diagram of the tar residue pyrolysis device.

### 2.3. Thermogravimetric Tests

Pyrolysis analysis was performed to determine the weight loss of TRP during the heating process. A thermal gravimetric analyzer (TA TGA55, TA Instruments, New Castle, DE, USA) was used to perform the TG experiment for pyrolysis. About 10 mg of tar residue was evenly spread on the bottom of the crucible at a heating rate of 25 °C/min from room temperature to 900 °C. The carrier gas (nitrogen) flow rate was 50 mL/min.

Non-isothermal TRP gasification experiments were performed using a thermal analyzer (Hitachi TG−DTA7200, Ibaraki, Japan). The carrier gas was $CO_2$ at a flow rate of 100 mL/min. TRP was heated from room temperature to 1000 °C at a rate of 20 °C/min and held for 10 min at 1000 °C.

### 2.4. Structural Characterization Methods

The samples were coated with Au before the experiment to make them conductive. The micromorphology of TRP was evaluated via scanning electron microscopy (SEM)

using a Gemini ZEISS 500 instrument (Jena, Germany). The SEM images were utilized to investigate the modifications in pore distribution induced by temperature variations.

The pore structure of the TRP was analyzed by nitrogen physisorption at 77 K. The specific surface area of TRP was calculated by the Brunauer–Emmett–Teller (BET) method (Micromeritics-ASAP2460, Norcross, GA, USA).

The TRP XRD data collection was performed by an X-ray diffractometer (Ultima IV, Tokyo, Japan) with a Cu target. The samples were packed onto a glass slide and scanned with a scanning speed of $2°/min$. Measurements were recorded from a start angle of $2\theta = 5°$ to an end angle of $80°$. The dimensions of the crystalline carbon expressed by the average diameter of the aromatic layers ($L_a$) and the average height of the stacked layers ($L_c$) can be obtained by the Debye-Scherrer equation (Equations (1) and (2)). The layer spacing ($d_{002}$) can be calculated using the Bragg equation (Equation (3)) [17]. The stacking layer numbers $N$ can be calculated using Equation (4)

$$L_a = 1.84\lambda/\beta_{100}cos\ \theta_{100} \tag{1}$$

$$L_c = 0.89\lambda/\beta_{002}cos\ \theta_{002} \tag{2}$$

$$d_{002} = \lambda/2sin\ \theta_{002} \tag{3}$$

$$N = L_c/d_{002} \tag{4}$$

where $\lambda$ ($\lambda = 0.15406$) represents the wavelength of X-rays, and $\beta$ and $\theta$ are the FWHM and peak position of the corresponding band, respectively.

Raman spectroscopy (ThermoFisher 2Xi, Waltham, MA, USA) was carried out to determine the carbon structural changes in TRP. Raman analysis was performed at room temperature and recorded in the range of 800 to 3500 $cm^{-1}$. Three points were detected for each sample in the Raman test. The difference between the three groups of data was not significant; therefore, the average of the assay data of the three points was used to ensure accuracy.

### 2.5. Experimental Data Processing of Non-Isothermal Gasification

The gasification carbon conversion ($x$) and gasification reaction rate ($r$) were obtained by the following equations [18]:

$$x = \frac{m_0 - m_t}{m_0 - m_{ash}} \tag{5}$$

$$r = \frac{dx}{dt} \tag{6}$$

where $m_0$ and $m_t$ represent the initial quality of the sample and the instantaneous weight at the gasification time of $t$, respectively, and $m_{ash}$ represents the final weight of ash remaining in TRP after being completely converted.

Several characteristic parameters were chosen to quantitatively depict the gasification synthesized from TRP prepared at distinct temperatures during the gasification procedure. The TG and DTG curves of TRP prepared at 500 °C are shown in Figure 2, which shows the definition of the initial gasification time ($t_i$) and the middle gasification time ($t_m$) of TRP.

Point A is selected to represent the point of the minimum value of the DTG curve, whose abscissa value is considered the middle gasification time ($t_m$). B is the intersection of the vertical line through point A and the TG curve. The tangent line is drawn at point B. The horizontal coordinate of the tangent line's intersection with $y = 1$ is the initial gasification time ($t_i$), and the horizontal coordinate of the tangent line's intersection with the horizontal line through the end of the TG curve is the final gasification time ($t_f$) [19,20].

The temperatures corresponding to the initial gasification time and the final gasification time are the initial gasification temperature ($T_i$) and the final gasification temperature ($T_f$).

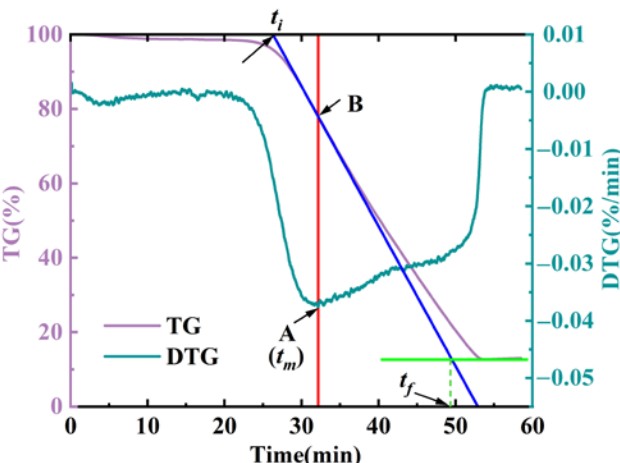

**Figure 2.** Schematic diagram for the gasification characteristic parameters.

To evaluate the comprehensive gasification reactivity of TRP, a comprehensive gasification characteristic index, $S$, was introduced for analysis. $S$ is calculated as the following equation [21]:

$$S = \frac{r_{max} \cdot r_{mean}}{T_i^2 \cdot T_f} \tag{7}$$

where $r_{max}$ and $r_{mean}$ represent the maximum and mean gasification rates, respectively.

## 3. Results and Discussion

### 3.1. Surface Structure Characteristics of the Pyrolysis Process

The properties of TRP were impacted by the pyrolysis temperature. Prior to characterizing the surface features, a brief analysis of the tar residue pyrolysis mechanism was conducted. As depicted in Figure 3, the pyrolysis process of tar residue can be divided into three key stages. The initial stage encompasses a temperature range of room temperature to 160 °C and involves moisture evaporation and desorption gases. The second stage occurs at temperatures ranging from 160 to 500 °C and consists of depolymerization and decomposition reactions. The TG curve rapidly decreases, with a mass loss of approximately 35%. The third stage, which occurs at temperatures above 500 °C, is primarily a condensation reaction, with a mass loss of approximately 33%. According to the pyrolysis process, the TRP prepared at 500, 600, 700, 800, and 900 °C, covering the third stage, was selected for characterization.

The micromorphology of TRP is shown in Figure 4, which includes the variation of the SEM image and BET-specific surface area with different pyrolysis temperatures.

As illustrated in Figure 4a–e, although pores were identified in TRP prepared at different pyrolysis temperatures, the density and size of the pores were different [22]. The pores of TRP prepared at 500 °C were sparse (Figure 4a), while TRP prepared at 600 °C exhibited denser but narrow pores (Figure 4b). It is evident from Figure 4c that the pores of TRP prepared at 700 °C are the largest of all samples. As the pyrolysis temperature rose, TRP synthesized at 800 °C displayed more dispersed pores (Figure 4d). In Figure 4e, the pore width of TRP prepared at 900 °C decreases. Therefore, we can conclude that the pyrolysis temperature significantly influences the distributions of pores in the TRP.

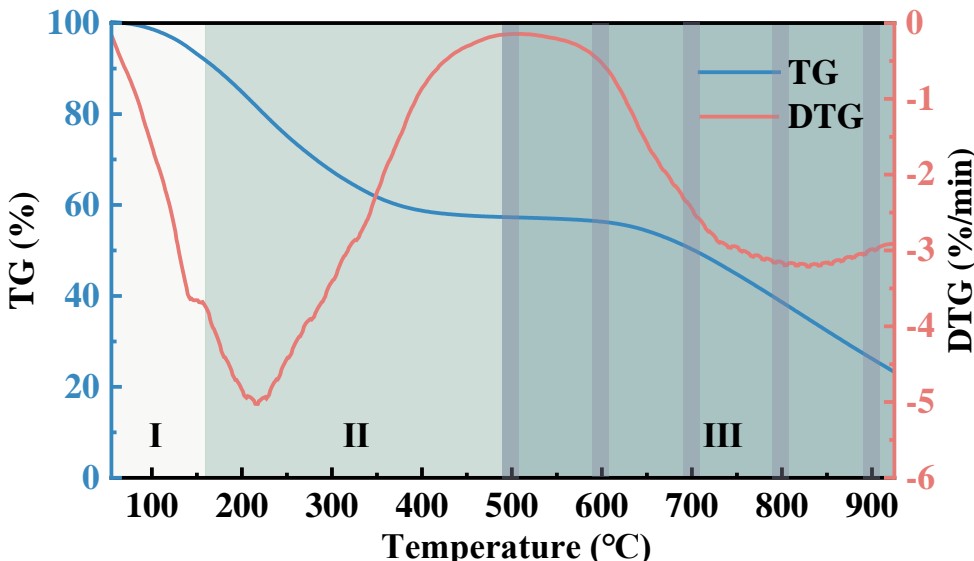

**Figure 3.** TG and DTG curves of tar residue.

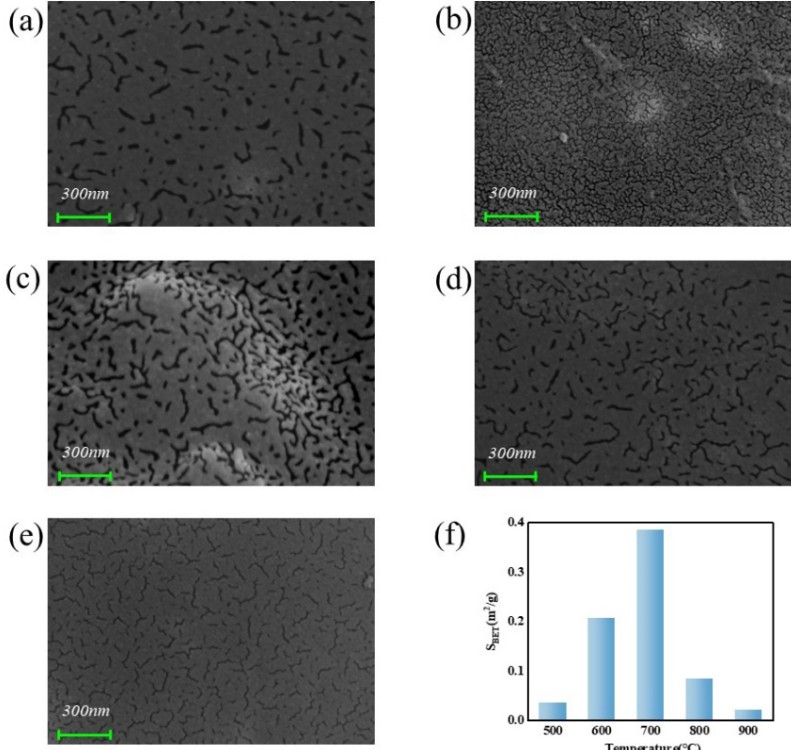

**Figure 4.** Evolution of the micromorphology of TRP prepared at different pyrolysis temperatures: (**a**) SEM-500 °C; (**b**) SEM-600 °C; (**c**) SEM-700 °C; (**d**) SEM-800 °C; (**e**) SEM-900 °C; (**f**) BET specific surface area.

The TRP SEM showed a trend of pores but not enough to support the evaluation of surface structure characteristics. The number of active sites on the surface of the sample significantly influences the level of difficulty in reacting between the gasification agent and carbon during the gasification reaction [23]. A large $S_{BET}$ generally implies a high number of active sites.

The effects of different pyrolysis temperatures on the BET-specific surface area of TRP are presented in Figure 4f. As manifested in Figure 4f, the $S_{BET}$ of TRP increased with an

increase in the pyrolysis temperature (in the range of 500–700 °C), with values ranging from 0.04 (at 500 °C) to 0.39 (at 700 °C). Within this temperature range, micromolecular and macromolecular organic matter dissociated and escaped from the particles. The rise in pyrolysis temperature favored the volatilization of gases and strengthened the development of macropores. Therefore, $S_{BET}$ gradually increased [24].

However, when the pyrolysis temperature increased from 800 to 900 °C, there was a decrease in $S_{BET}$. As the pyrolysis temperature increased, the tar generated from tar residue pyrolysis had a secondary reaction. The product of the secondary reaction was deposited in the pores on the surface of the sample, causing some of the pores to be jammed [25–27]. Therefore, pores were reduced or became narrower, and the $S_{BET}$ decreased.

### 3.2. Carbon Structural Characteristics of the Pyrolysis Process

3.2.1. Carbon Structural Characteristics Obtained from XRD Analysis

X-ray diffraction (XRD) is a technique utilized for investigating structural characteristics, which provides information on microcrystalline parameters. Figure 5 illustrates the XRD pattern variations of TRP obtained through diverse pyrolysis degrees. TRP synthesized at different pyrolysis temperatures exhibited two distinct peaks at 15–30° and 40–50°, respectively.

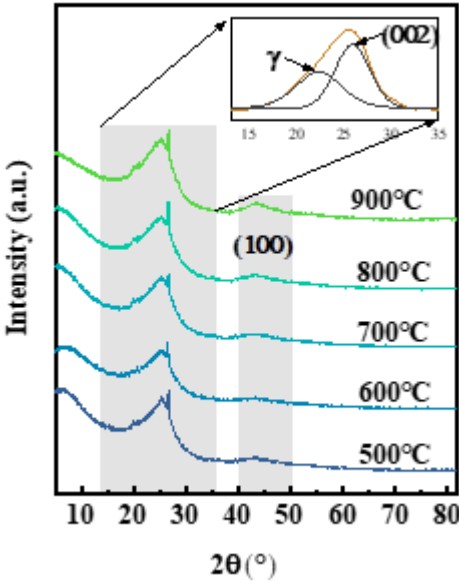

**Figure 5.** Comparison of TRP XRD patterns at different pyrolysis temperatures.

Many studies [28,29] have shown that the peaks present from 15° to 30° are constituted by the combined superposition of the 002-peak and the γ-peak, which is to the left of the 002-peak. The 002-peak is the stacking between the structures of the aromatic layer. The γ-peak is related to the structure of the side chains of aliphatic hydrocarbons, various functional groups, and alicyclic hydrocarbons connected to the structure of the aromatic rings, and the 100-peak represents the size of the aromatic layer.

The XRD data were fitted using PeakFit software to calculate the microcrystalline structure parameters of the TRP, and the results are shown in Table 2.

For TRP prepared at different pyrolysis temperatures, $d_{002}$ decreased slightly from 3.457 to 3.435 Å with increasing pyrolysis temperature. However, compared to the layer spacing of graphite of 3.354 Å, the TRP layer spacing was larger. With increasing pyrolysis temperature, the aromatic layers experienced condensation, leading to a decline in $d_{002}$. However, the TRP microcrystals remained less structured and were not fully graphitized within this temperature range.

**Table 2.** Structural parameters of TRP at various pyrolysis temperatures.

| Pyrolysis Temperatures/(°C) | $d_{002}$/(Å) | $L_a$/(Å) | $L_c$/(Å) | $N$ |
|---|---|---|---|---|
| 500 | 3.457 | 23.75 | 16.37 | 4.74 |
| 600 | 3.456 | 25.42 | 17.57 | 5.08 |
| 700 | 3.455 | 26.40 | 18.12 | 5.24 |
| 800 | 3.445 | 28.57 | 18.92 | 5.49 |
| 900 | 3.435 | 34.99 | 20.72 | 6.03 |

$L_a$ and $L_c$ reflected the crystallite size, which ranged from 23.75–34.99 and 16.37–20.72 Å, respectively. Meanwhile, the number of stacking layers $N$ increased from 4.74 to 6.03 accordingly. Therefore, the crystallite grew during the pyrolysis process. Higher temperatures (within 500–900 °C) favor the formation of the bigger aromatic nucleus in TRP because higher temperatures may reduce interfacial defects between adjacent and the occurrence of condensation reaction increases the size of aromatic nucleus.

### 3.2.2. Carbon Structural Characteristics Obtained from Raman Spectroscopy Analysis

Raman spectroscopy can be used to analyze the structural evolution of solid carbonaceous materials [30]. Figure 6 shows the Raman spectra of the TRP after baseline correction.

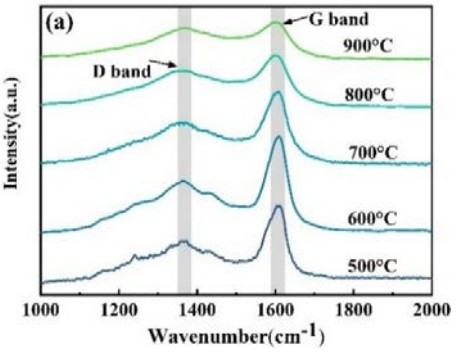
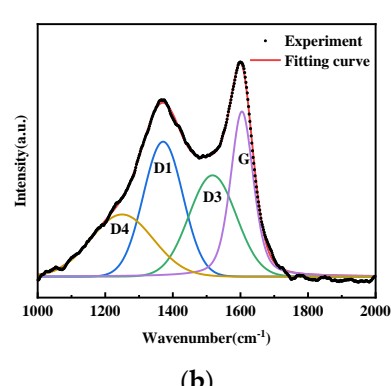

**Figure 6.** Raman spectra of TRP: (**a**) normalized intensity; (**b**) imitating results.

TRP spectra prepared at different pyrolysis temperatures typically exhibit two overlapping peaks, namely, peak D at 1100–1500 cm$^{-1}$ and peak G at 1500–1700 cm$^{-1}$ [31]. However, the original Raman spectra are the result of the superposition of several peaks, which need to be further split and adjusted into five peaks—D1, D2, D3, D4, and G. Peak D1 in the 1350 cm$^{-1}$ band commonly refers to vibrations related to disordered graphite lattices with imperfections in the plane, such as defects and heteroatoms. The presence of D3 and D4 indicates that there are a large number of amorphous forms of carbon in the TRP, reflecting active sites. Peak G is related to an ideal graphite lattice vibrational mode with $E_{2g}$ symmetry [32–34]. During the fitting process, it was found that the fitting of peak D2 was relatively low and the position was far away, so four peaks were used to fit the Raman spectra of the pyrolysis product (D1, D3, D4, and G) [35].

Accordingly, both the ratio of peak G intensity to all peak intensities ($I_G/I_{All}$) and the ratio of peak D1 intensity to peak G ($I_{D1}/I_G$) intensity represent the graphitization degree to some extent [33,36]. More precisely, the ordering of the carbon structure increased with increasing $I_G/I_{All}$ and decreased with increasing $I_{D1}/I_G$ [37]. Peak D (consisting of peak D1, peak D3, and peak D4) could be viewed as a disordered structure in TRP and was more reactive than peak G. Peak D1 denoted the microcrystalline structure in the disordered configuration, while peak D3 and peak D4 signified amorphous structures that were usually more readily transformed into and consumed by active structures [34,38]. Thus, the ratio of

the sum of peak D3 and peak D4 intensity to D peak intensity ($I_{D3 + D4}/I_D$) could evaluate the structure of amorphous carbon.

Parameters such as the intensity ratio $I_G/I_{All}$, $I_{D1}/I_G$, and $I_{D3 + D4}/I_D$ are used to evaluate the degree of order of carbon structure and carbon activity, as shown in Figure 7 [39].

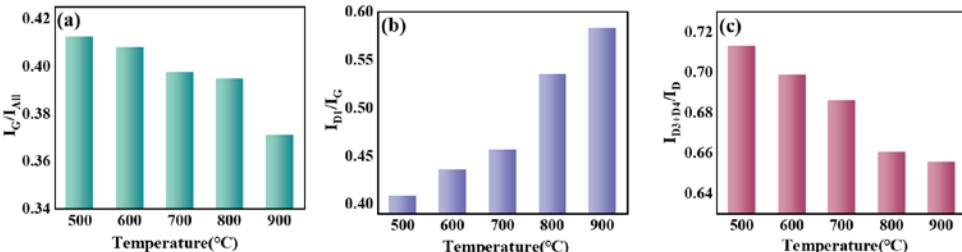

**Figure 7.** TRP intensity ratios prepared at different temperatures: (**a**) $I_G/I_{All}$; (**b**) $I_{D1}/I_G$; (**c**) $I_{D3 + D4}/I_D$.

The increase in the pyrolysis temperature from 500 to 900 °C resulted in an increase in $I_{D1}/I_G$ and a decrease in $I_G/I_{All}$, indicating that the carbon structure of TRP had become disorderly and the ordered graphite carbon structure had been reduced, as shown in Figure 7a,b. The primary pyrolysis reaction of tar residue involved condensation above 500 °C. However, since the pyrolysis temperature was below 1000 °C, the units of the aromatic structure formed by the condensation reaction were comprised of no more than 10–12 condensed aromatic rings with a lower degree of stacking [40]. Although the size and concentration of the aromatic structure increased under such pyrolysis conditions, the temperature was insufficient to form a large aromatic nucleus, let alone completely graphitize them [41]. It was observed that when the amount of microcrystalline structure increased, the proportion of defect structure also increased, which led to the increase in the value of $I_{D1}/I_G$.

As shown in Figure 7c, $I_{D3 + D4}/I_D$ decreases with increasing pyrolysis temperature. As the pyrolysis reaction proceeded, the active structures in TRP were gradually reduced, including the consumption of oxygen-containing structures, the decomposition of substitution groups, and the breakage of cross-linked structures. Meanwhile, the tar residue caused the condensation reaction, and the aromatic structure of the small ring was transformed into the large ring structure [38]. All of these led to a decrease in $I_{D3 + D4}/I_D$.

### 3.3. Non-Isothermal Gasification Reaction and Reactivity Analysis

The non-isothermal gasification TG and DTG curves of TRP under a $CO_2$ atmosphere are indicated in Figure 8. It is evident that the TG and DTG curves of TRP synthesized at distinct pyrolysis temperatures exhibit analogous features. The TRP gasification procedure can be categorized into three phases: preheating, rapid gasification reaction, and termination. The second stage is the most reactive stage, and the gasification characteristics in this stage can well represent the gasification reaction characteristics of TRP. Therefore, the analysis of gasification reactivity was focused mainly on this stage [42].

The parameters of the gasification characteristics of TRP prepared at different pyrolysis temperatures are shown in Table 3. It can be found that the value of $t_i$ continuously increased with the pyrolysis temperature of the prepared TRP. This indicates that the gasification reaction of TRP prepared at lower pyrolysis temperatures begin at an earlier gasification time.

However, the variation pattern of $t_m$ at different degrees of pyrolysis was not the same as that of $t_i$. With increasing temperature (500–700 °C), $t_m$ gradually decreased. When the temperature increased above 800 °C, $t_m$ had an obvious tendency to increase. A minimum value occurred at a pyrolysis temperature of 700 °C. This indicates that TRP synthesized at 700 °C attained the midway point of the gasification reaction sooner than TRP prepared at other temperatures. The low values of $t_m$ suggested that the gasification reactivity of TRP

would be elevated. This means that the gasification reactivity of TRP was the highest at a pyrolysis temperature of 700 °C.

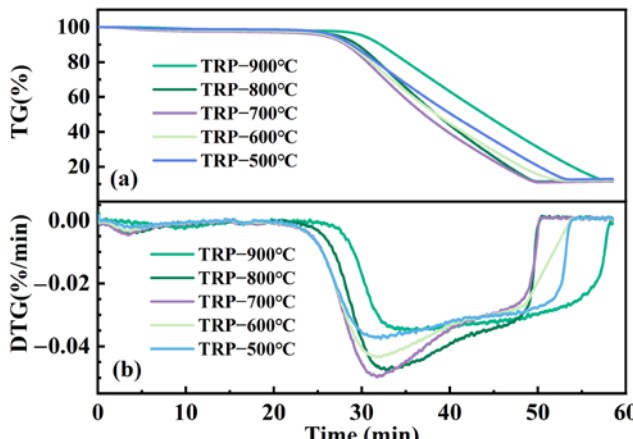

**Figure 8.** The non-isothermal gasification TG and DTG curves of TRP: (**a**) TG; (**b**) DTG.

**Table 3.** Gasification characteristics parameters of TRP prepared at different pyrolysis temperatures.

| Pyrolysis Temperature/°C | $t_i$/min | $t_m$/min | $S \times 10^{-12}$/min$^{-2}\cdot$°C$^{-3}$ |
|---|---|---|---|
| 500 | 26.32 | 32.15 | 2.06 |
| 600 | 26.47 | 32.03 | 2.41 |
| 700 | 26.60 | 31.68 | 2.72 |
| 800 | 27.65 | 32.80 | 2.39 |
| 900 | 29.36 | 35.91 | 1.58 |

It can also be found in Table 3 that when the pyrolysis temperature was in the range of 500 to 700 °C, the comprehensive gasification characteristics index $S$ increased from 2.06 to 2.72 with increasing pyrolysis temperature. However, when the pyrolysis temperature increased to within the 800–900 °C range, the $S$ value decreased from 2.39 to 1.58 with increasing pyrolysis temperature. $S$ is used to more accurately represent the gasification reactivities of TRP prepared at different pyrolysis temperatures [43]. This observation indicates that the increase in the degree of pyrolysis (from 500 to 700 °C) favored an increase in the gasification reactivity of TRP. Worse results were observed when increasing the pyrolysis temperature further, to above 800 °C, in which the gasification reactivity was reduced with increasing pyrolysis temperature. Consequently, TRP prepared at 700 °C had the best gasification reactivity.

The pyrolysis temperature would significantly affect the surface structure and carbon structure of TRP, and the structure variation would also influence the magnitude of the gasification reactivity. To further investigate the dominant relationship between gasification reactivity and structure, the parameters of surface structure and carbon structure were correlated with parameters of gasification characteristics (seen in Figures 9–11).

Figure 9 shows the relationships between the initial gasification time and the structure parameters. As can be seen in Figure 9, $t_i$ is inversely proportional to $I_{D3+D4}/I_D$ and positively proportional to $N$, while there is no significant functional relationship with $S_{BET}$. The carbon structure ($I_{D3+D4}/I_D$ and $N$) was more important for $t_i$ than the surface structure ($S_{BET}$), particularly for $I_{D3+D4}/I_D$, which may reflect the amorphous carbon trend in TRP.

During the initial stage of gasification, $CO_2$ showed a preference for reacting with amorphous carbon. The $N$ parameter can be utilized to evaluate the degree of stacking in the aromatic nucleus, where higher $N$ values correspond to a more stable aromatic structure. Accordingly, TRP (prepared at 500 °C) with more amorphous carbon and a lower stacking degree would generally have an earlier initial gasification time.

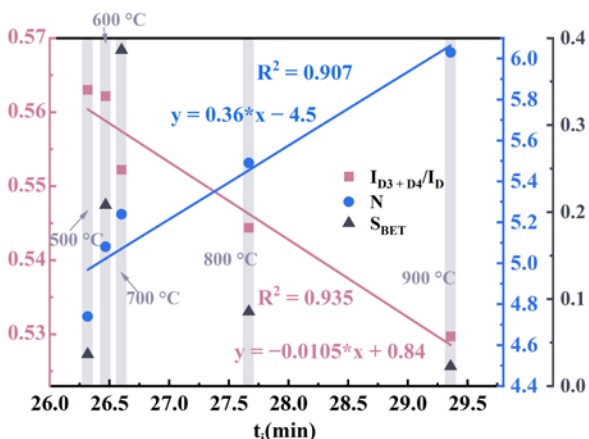

**Figure 9.** Relationship between $t_i$ and structure parameters.

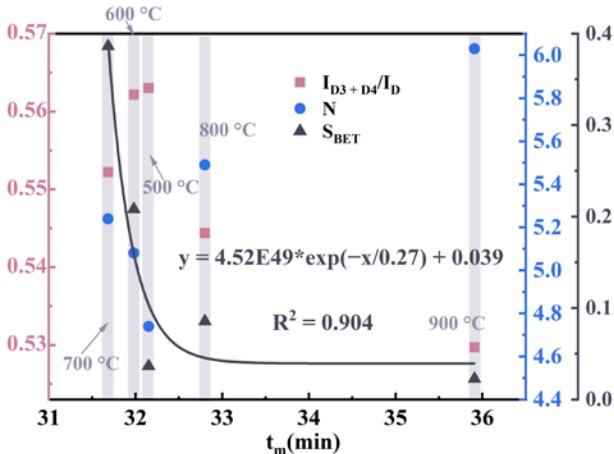

**Figure 10.** Relationship between $t_m$ and structure parameters.

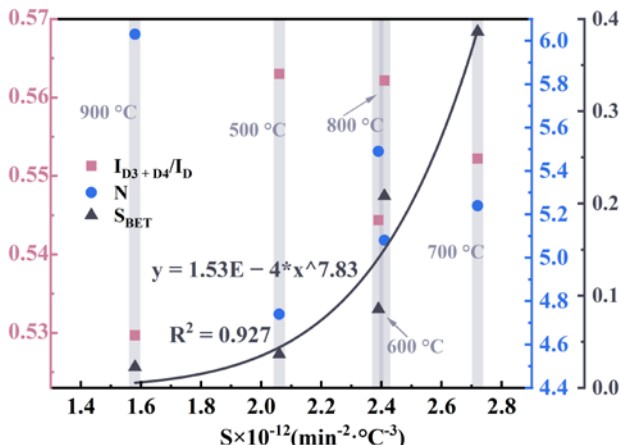

**Figure 11.** Relationship between S and the structure parameters.

As can be seen in Figure 10, a wonderful exponent correlation between $S_{BET}$ and $t_m$ of $S_{BET} = 4.52 \times 10^{49} e^{(-t_m/0.27)} + 0.039$, with $R^2 = 0.904$ was observed, while the correlation between $I_{D3+D4}/I_D$ and $N$ with $t_m$ was relatively low. The correlation between the surface structure and $t_m$ was greater than that of the carbon structure. A larger $S_{BET}$ corresponds to an earlier middle gasification time. This is explained by the fact that larger $S_{BET}$ and pores possibly resulted in the easier entry of the gasification agent into the

macromolecular structure of TRP and an increased effective contact area between the TRP and the gasification agent, leading to an increase in the chance of chemical reactions.

Figure 11 shows the relationships between $S$ and the structure parameters. There was a clear functional relationship between $S_{BET}$ and $S$ of $S_{BET} = 1.53 \times 10^{-4}S^{7.83}$, with $R^2 = 0.927$, while $I_{D3+D4}/I_D$ and $N$ did not have an obvious functional relationship with $S$. For $S$, the surface structure played a more important role than the carbon structure. As a result of the fine correlation between S and $S_{BET}$, the $S_{BET}$ could act as a rough indicator for evaluating the gasification reactivity of TRP.

During the gasification of TRP from room temperature to 1000 °C, distinct structural parameters predominantly affected the characteristic parameters of various gasification stages. This suggests that the gasification reaction rate is regulated by diverse factors at different phases. In the initial gasification phase, the reaction rate was primarily influenced by the carbon structure due to the ample active sites present on the surface of TRP particles. The more amorphous carbon and the fewer the stacking layers of the aromatic nucleus, the earlier the reaction occurs. Further, as the gasification reaction proceeded, the active sites on the surface of the TRP particle were gradually consumed, and the gas–solid reaction between TRP and $CO_2$ was mainly influenced by the diffusion rate of the $CO_2$ gas. The larger surface area benefited the gasification reaction process. Therefore, a single physicochemical structure parameter alone cannot accurately evaluate all the gasification characteristic parameters.

## 4. Conclusions

This study has discussed the effect of the pyrolysis temperature (500–900 °C) on the structural characteristics and gasification reactivity of TRP. An in-depth analysis was conducted to examine the relationship between structural parameters and diverse gasification characteristic parameters. Based on the acquired outcomes, the following conclusions were drawn.

(1) TRP prepared at different pyrolysis temperatures can be divided into two stages in terms of surface structure, namely, the development of the pore stage in the range of 500–700 °C and the pore-jamming stage in the range of 800–900 °C. Among all the samples, TRP prepared at 700 °C had the best pore structure and specific surface area ($S_{BET}$). Adding such TRP to the rotary bottom furnace can provide a more sufficient reduction in volume and increase the reaction rate.

(2) When the pyrolysis temperature increased (500 to 900 °C), the TRP was subject to condensation but not graphitization. $L_a$, $L_c$, and $N$ increased and $I_{D3+D4}/I_D$ decreased with temperature, representing that TRP, the aromatic structure of the small ring structure, transformed into the large ring structure.

(3) In the gasification reaction stage of TRP, $t_i$ (initial gasification time) increased with increasing pyrolysis temperature. The chemical structure ($I_{D3+D4}/I_D$ and $N$) was more relevant for $t_i$ than the physical structure ($S_{BET}$). $t_m$ (middle gasification time) had a good exponential correlation with $S_{BET}$, and the higher $S_{BET}$ (at pyrolysis temperature of 700 °C) corresponded to the earlier middle gasification time. $S$ (comprehensive gasification characteristic index) first increased and then decreased as the pyrolysis temperature increased, reaching a maximum of 700 °C. There was a good correlation between $S$ and $S_{BET}$.

(4) The $CO_2$ gasification of TRP was investigated, wherein the carbon structure exerted a greater influence during the initial phase of the gasification reaction; the carbon structure played a more important role than the surface structure. As the gasification reaction progressed, the active sites on the TRP particle surface decreased, resulting in a stronger association between the surface structure and the gasification reaction. TRP prepared at a pyrolysis temperature of 700 °C had the best gasification reactivity.

**Author Contributions:** J.L. performed the experiments and analyzed the data; W.L. performed part of the experimental data collection; J.L. drafted the paper; X.S. and J.S. polished the manuscript; P.L. contributed reagents/materials; Q.X. advised in the process of writing the manuscript. All authors have read and agreed to the published version of the manuscript.

**Funding:** This research was funded by the National Key R&D Program of China (grant number 2019YFC1905703) and the Provincial Science and Technology Plan Projects in Guangdong Province (grant number GDKJ2020002).

**Data Availability Statement:** The datasets generated and analyzed during the current study are available from the corresponding author upon reasonable request.

**Conflicts of Interest:** The authors declare no conflict of interest.

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
