# Peer review of "The Correlation between the Structure Characteristics and Gasification Characteristics of Tar Residue from Pyrolysis"

_sustainability, doi:10.3390/su15097130_

Round 1

Reviewer 1 Report

The work portrays a study of the operating conditions for pyrolysis reaction. Despite showing some scientific merit and contributing to the advancement of science, this type of work proves to be repetitive in the literature for this reaction. Some modifications are suggested to be ready to be published in the journal:

1) The references do not appear in the document (formatting error), making it impossible to analyze the works cited.

2) Coke production may be indicated, but for global values (around the world) and not for a concrete case study (as in China).

3) Put a reference to support this sentence "...Au before the experiment to make them conductive.".

4) Put variables in italic; for instance: "...time of t,...". 

5) Please, indicate the temperatures that produce the material with the most suitable pores for the applications mentioned in the introduction.

6) The work must indicate the main purpose of this study for the development of technology and society.

Author Response

The work portrays a study of the operating conditions for pyrolysis reaction. Despite showing some scientific merit and contributing to the advancement of science, this type of work proves to be repetitive in the literature for this reaction.

ANSWER: Thank you for your comments. We have carefully revised· the whole manuscript according to your comments.

Q1. The references do not appear in the document (formatting error), making it impossible to analyze the works cited.

ANSWER: In our resubmitted manuscript, the formatting errors of the references is revised. We were really sorry for our careless mistakes. Thank you for your reminder.

Q2. Coke production may be indicated, but for global values (around the world) and not for a concrete case study (as in China).

ANSWER: We think this is an excellent suggestion. According to your suggestion, we have added the global coke production in the introduction section, and explained that China accounts for more than half of the global coke production, as well as other countries that generate more coke. (L36-42)

Q3. Put a reference to support this sentence "...Au before the experiment to make them conductive.".

ANSWER: Thanks for your professional suggestion. We have made corresponding modifications based on your comments. (L143)

Q4 Put variables in italic; for instance: "...time of t,...".

ANSWER: We have modified the format of the arguments and dependent variables appearing in the article. Thanks for your suggestions.

Q5. Please, indicate the temperatures that produce the material with the most suitable pores for the applications mentioned in the introduction.

ANSWER: The TRP prepared at 700 °C has the best pores structure and SBET among all samples. Pyrolysis temperature of 700 °C is the most suitable porosity of the material produced for subsequent addition as a reducing agent to the rotary bottom furnace. This explanation has been added to conclusion (1). (L436-439)

Q6. The work must indicate the main purpose of this study for the development of technology and society.

ANSWER: I think that the study of pyrolysis of tar residue for subsequent gasification is of great significance in various aspects. This not only can alleviate the energy crisis, reduce the dependence on traditional fossil fuels, but also improve energy utilization efficiency and reduce negative environmental impact. Meanwhile, in the process of implementing this technology, it will also create employment opportunities and promote local economic development. In addition, the technology and process involved in the study of thermal decomposition and subsequent gasification of tar slag also provide a basis for the treatment of other energy-containing solid wastes in the metallurgical industry, promoting sustainable utilization modes of energy and resources. (L75-82)

Reviewer 2 Report

The explanations of the authors are sufficient and the material is well presented. The display of the device used has also helped to convey the content.

Examining and studying this work has made me interested in the present work

• What is the main question addressed by the research?
In this work tar residue from slow pyrolysis was prepared from tar residue at different pyrolysis temperatures and characterized by various methods.
• Do you consider the topic original or relevant in the field? Does it address a specific gap in the field?
Yes, this topic is original or relevant in the field. The importance of the subject has not been fully discussed in the work.
• What does it add to the subject area compared with other published material?
The parameters examined in this work and the work style are completely different from previous works.
• What specific improvements should the authors consider regarding the methodology? What further controls should be considered?
The items discussed in the work are appropriate.
• Are the conclusions consistent with the evidence and arguments presented and do they address the main question posed?
Yes, the conclusions are consistent with the evidence and arguments presented. The authors address the main question posed.
• Are the references appropriate?
Yes, the references are appropriate. However, references are not clear in the text.
• Please include any additional comments on the tables and figures.
The SEM figures are not clear.
Line 103: Why was a heating rate of 7 °C/min applied? What about N2 flow rate? On what basis were these temperature and flow ranges used? Has optimization been done? 

Author Response

The explanations of the authors are sufficient and the material is well presented. The display of the device used has also helped to convey the content.

ANSWER: Thank you for your positive comments. Regarding the issue of English language and style, we tried our best to improve the manuscript and made some changes to the manuscript. These changes will not influence the content and framework of the paper. And here we did not list the changes but marked in red in the revised paper. We appreciate for Reviewers’ warm work earnestly and hope that the correction will meet with approval.

Q1. The importance of the subject has not been fully discussed in the work.

ANSWER: Thank you for your reminder. The importance of this research has been added to the introduction of the article. This not only can alleviate the energy crisis, reduce the dependence on traditional fossil fuels, but also improve energy utilization efficiency and reduce negative environmental impact. Meanwhile, in the process of implementing this technology, it will also create employment opportunities and promote local economic development. In addition, the technology and process involved in the study of thermal decomposition and subsequent gasification of tar slag also provide a basis for the treatment of other energy-containing solid wastes in the metallurgical industry, promoting sustainable utilization modes of energy and resources. (L75-82)

Q2. However, references are not clear in the text.

ANSWER: The references have been modified. We were really sorry for our careless mistakes. Thank you for your reminder.

Q3. The SEM figures are not clear.

ANSWER: Corresponding adjustments have been made to Figure 4(c) and (d). (L220)

Q4. Line 103: Why was a heating rate of 7 C/min applied? What about N2 flow rate? On what basis were these temperature and flow ranges used? Has optimization been done?

ANSWER: In the industrial experimental stage of subsequent pyrolysis, we may use a multi-chamber furnace for to pyrolysis the tar residue. The temperature of each layer in the multi-chamber furnace is different, ranging from 300℃ to 900℃, with the temperature increasing from low to high from top to bottom. The heating rate of raw materials entering the furnace from the top is about 7 ℃/min. In order to better study the optimal state of tar residue pyrolysis in multi-chamber furnace, the pyrolysis temperature of tar residue in the laboratory was set at 7 ℃/min.

Before the experiment began, N2 was used to purge the quartz tube, ensuring that the sample was heated in an inert gas atmosphere. During the experiment, because the quartz tube is sealed, N2 is mainly used to remove some of the gas generated during the pyrolysis process and maintain a relatively stable reaction environment. A nitrogen flow rate of 50 mL/min is sufficient to complete these tasks without causing waste.

Reviewer 3 Report

In the manuscript sustainability-2258378  entitled “The correlation between Structure Characteristics and Gasification Characteristics of Tar Residue from pyrolysis,” authors evaluated the variation patterns of microscopic morphology, surface area, and carbon structure characteristics of TRPP with pyrolysis temperature to illustrate the effect of the pyrolysis temperature on the CO2 gasification characteristics parameters of TRP—the correlation between surface structure, carbon structure characteristics, and gasification parameters at different stages.

Figure 1 is not precise.

Figure 2, ti and tf must be recorded.

Is Figure 3 a result of the current work? If yes, it is poor in quality. If not, it needs a reference.

The scale of Figure 4 C&D  is not available.

Line 30, reference is missed.

Line 36, reference is missed.

Line 36, reference is missed

Line 37, reference is missed

 Line 38, reference is missed

Line 39, reference is missed

Line 40, reference is missed

…..

Equations 1-4 are mistyped

In figure 6b, deconvolution is poor; another peak at more than 1600 is required to capture the experimental data.

Figure 10, try to plot 1/x and make data linear.

Figure 11, try to plot log x and make data linear.

Author Response

In the manuscript sustainability-2258378  entitled “The correlation between Structure Characteristics and Gasification Characteristics of Tar Residue from pyrolysis,” authors evaluated the variation patterns of microscopic morphology, surface area, and carbon structure characteristics of TRPP with pyrolysis temperature to illustrate the effect of the pyrolysis temperature on the CO2 gasification characteristics parameters of TRP—the correlation between surface structure, carbon structure characteristics, and gasification parameters at different stages.

ANSWER: Thank you for your positive comments. We have carefully revised the whole manuscript according to your comments. Regarding the issue of English language, we tried our best to improve the manuscript and made some changes to the manuscript. These changes will not influence the content and framework of the paper. And here we did not list the changes but marked in red in the revised paper. We appreciate for Reviewers’ warm work earnestly and hope that the correction will meet with approval.

Q1. Figure 1 is not precise.

ANSWER: Thanks for your suggestion. We have redrawn this part according to the Reviewer’s suggestion. (L127)

Q2. Figure 2, ti and tf must be recorded.

ANSWER: Thanks for your suggestion. ti and tf have been clearly marked in Figure 2. (L184)

Q3. Is Figure 3 a result of the current work? If yes, it is poor in quality. If not, it needs a reference.

ANSWER: Thanks for your suggestion. Figure 3 is a result of the current work, and the color and quality of the image have been modified. (L215)

Q4. The scale of Figure 4 C&D is not available.

ANSWER: Thanks for your suggestion. Corresponding adjustments have been made to Figure 4(c) and (d). (L220)

Q5. Line 30, reference is missed….

ANSWER: In our resubmitted manuscript, the formatting errors of the references is revised. We were really sorry for our careless mistakes. Thank you for your reminder.

Q6. Equations 1-4 are mistyped

ANSWER: Thank you for your reminder. We have modified Equations 1-4.

Q7. In figure 6b, deconvolution is poor; another peak at more than 1600 is required to capture the experimental data.

ANSWER: Thank you for your suggestion. We moved the position of the D3 peak in Figure 6 (b) to make the fitting better, and made corresponding modifications to both Figure 6 (b) and Figure 7. Regarding another peak at more than 1600, we have tried many times, but we cannot obtain a better fit. Either it completely coincides with the G peak, or the peak position is relatively distant.

Q8. Figure 10, try to plot 1/x and make data linear. Figure 11, try to plot log x and make data linear.

ANSWER: We tried to plot with 1/x and logx. However, it may be because the horizontal and vertical coordinate differences are too large, and this function cannot show a trend that can be fitted within this range.

Round 2

Reviewer 1 Report

All ok.

Reviewer 3 Report

It is OK to be published.